# Post-Hurricane Distress Scale (PHDS): Determination of General and Disorder-Specific Cutoff Scores

**DOI:** 10.3390/ijerph19095204

**Published:** 2022-04-25

**Authors:** Yonatan Carl, Andy Vega, Gina Cardona-Acevedo, Marina Stukova, Melissa Matos-Rivera, Anamaris Torres-Sanchez, Melissa Milián-Rodríguez, Brian Torres-Mercado, Grisel Burgos, Raymond L. Tremblay

**Affiliations:** 1San Juan Bautista School of Medicine, Caguas 00726, Puerto Rico; melissamr@sanjuanbautista.edu (M.M.-R.); anamarists@sanjuanbautista.edu (A.T.-S.); melissa.milian@sanjuanbautista.edu (M.M.-R.); brianjtm@sanjuanbautista.edu (B.T.-M.); gburgos@sanjuanbautista.edu (G.B.); raymond.tremblay@sanjuanbautista.edu (R.L.T.); 2Medical College of Wisconsin, Milwaukee, WI 53226, USA; anvega@mcw.edu; 3Universidad Central del Caribe, Bayamón 00960, Puerto Rico; 118gcardona@uccaribe.edu; 4University of Miami, Coral Gables, FL 33146, USA; mxs7873@med.miami.edu

**Keywords:** mental health, PHDS, cutoff, depression, anxiety, PTSD, hurricane, disaster medicine

## Abstract

The Post-Hurricane Distress Scale (PHDS) was developed to assess mental health risk in the aftermath of hurricanes. We derive both disorder-specific cutoff values and a single nonspecific cutoff for the PHDS for field use by disaster relief and mental health workers. Data from 672 adult residents of Puerto Rico, sampled 3 to 12 months after Hurricane Maria, were collected. Participants completed a five-tool questionnaire packet: PHDS, Kessler K6, Patient Health Questionnaire 9, Generalized Anxiety Disorder 7, and Post-Traumatic Stress Disorder Checklist for DSM V (PCL-5). ROC curves, AUC values, sensitivities, specificities, Youden’s index, and LR+ ratios are reported. The recommended single cutoff value for the PHDS is 41, whereby a respondent with a PHDS score of 41 or above is deemed high-risk for a mental health disorder. The single field use PHDS cutoff demonstrated high specificity (0.80), an LR + ratio (2.84), and a sensitivity of 0.56. The mean ROC values of PHDS for Kessler K6, Patient Health Questionnaire 9, Generalized Anxiety Disorder 7, and PCL-5 were all above 0.74. The derived cutoff for the PHDS allows efficient assessment of respondents’ and/or a community’s risk status for mental health disorders in the aftermath of hurricanes and natural disasters.

## 1. Introduction

Disaster-affected populations have a higher prevalence of mental health problems than that of the general population [1,2,3]. According to Galea and Kopala-Sibley [4,5], after hurricanes and other natural disasters the prevalence of post-traumatic psychological disorders is around 34%, while other researchers state that overall prevalence ranges from 5% to 50% [2,4,6,7,8]. Some of the psychological outcomes reported in populations after large-scale traumatic events include post-traumatic stress disorder (PTSD), acute stress disorder (ASD), depression, and generalized anxiety disorder (GAD) [1,9]. Multiple factors correlate with these psychological outcomes and can be classified as pre-, peri- and/or post-disaster risk factors [10,11]. Pre-disaster risk factors include gender, age, and prior mental illness (PMI), while peri-disaster risk factors include severity and grade of exposure, injury, loss of loved ones, and loss of home and/or possessions. Two post-disaster risk factors are prolonged stressors and a lack of social support [2,9,12,13,14,15]. While exposure to an actual traumatic event is strongly correlated with psychological outcomes, secondary and/or prolonged stressors, such as post-traumatic events related to living conditions and access to resources and medical services, are also significant contributors [16,17]. Measuring the impact of both acute and prolonged stressors in the aftermath of disasters is relevant when assessing the risk of developing mental health disorders [15,18,19].

The Post-Hurricane Distress Scale (PHDS) questionnaire was developed and validated in the aftermath of Hurricane Maria [12]. Designed to help relief workers and researchers assess a populations’ distress and emotional trauma, it quantifies both acute and prolonged post-disaster stressors. When compared with other traumatic exposure inventories, such as the Traumatic Event Screening Scale (TESS) and the Hurricane-Related Traumatic Experience (HURTE), the PHDS questionnaire was shown to have higher specificity and better performance in a post-hurricane setting, due to its inclusion of prolonged post-disaster stressors [12]. The PHDS assesses acute peri-traumatic and post-traumatic stressors and is not limited to hurricanes [12,20]. The approximately 5-min questionnaire’s score can be used as a comparative measure of post-disaster distress but, as it does not have a validated cutoff value, it cannot identify individuals at risk for developing disaster-related mental health disorders. The original study’s aim was to develop an instrument—and validate its use—for measuring an individual’s and population stressors and to compare it with other existing traumatic exposure inventories.

The current study’s goal is to determine a cutoff threshold value that best correlates PHDS scores with validated mental health risk assessment instruments, including the Kessler K6 (K6), the Patient Health Questionnaire 9 (PHQ-9), the Generalized Anxiety Disorder 7 (GAD-7), and the PTSD Checklist for the DSM-V(PCL-5). This will allow researchers and relief workers to use the PHDS to identify communities and individuals that are at risk of developing psychiatric disorders, both immediately after the disaster and throughout the recovery process, as well as identifying community need for mental health relief efforts.

## 2. Materials and Methods

### 2.1. Sampling

Over the course of this study, a total of 672 participants across Puerto Rico (PR) completed a demographic survey and a mental health questionnaire packet. Written informed consent was obtained from all participants and kept separately from the completed questionnaires. Participants were selected using a multistage sampling of adults, from the San Juan Region, Ponce, and Caguas—the largest cities of PR—in addition to 17 smaller municipalities across PR. Within any single municipality, a sample of private households was chosen at random. Participation was limited to 1 participant, of at least 18 years of age, per household. A complete list of the sampling sites is included in Table 1. Data collection began 3 months after the hurricane and continued until 12 months after the hurricane. The demographic variables of the population sampled include age, gender, civil status, race/ethnicity, education level, employment status, combined household income, household ownership, and current living location (Table 1).

We chose not to exclude any participants on the grounds of self-reported pre-existing mental health conditions as they represent a valid segment of any post-disaster population and thus should be included in the derivation of a cutoff in a post-disaster population. Relocation, whether temporary or permanent, is not included in any of the four factors that comprise the PHDS [12]. A parallel study by our group, currently under review, does assess the contribution of homelessness and/or relocation to mental health outcomes; this study does not make a distinction of respondents based on having relocated in the aftermath of the storm.

All participants were given the purpose of the study by one of the researchers and, if consent was given, the participant was provided with a written consent form that was completed separately to ensure anonymity. Questionnaires were self-administered in the presence of a trained researcher. If, at any point, a participant requested termination of the session, due to emotional distress, the session was ended and participants were informed of psychological resources available to them either through the San Juan Bautista School of Medicine or others available island-wide. This administration protocol was reviewed by the San Juan Bautista School of Medicine (SJBSOM) Institutional Review Board and approved as protocol #22-2018.

### 2.2. Assessment Tools

#### 2.2.1. PHDS Questionnaire

The PHDS is a 20-item 4-factor (Being in Need, Resource Loss, Personal Safety, and Health Concerns) self-administered instrument, available in both English and Spanish, developed for use in the field during the recovery period after hydrological and meteorological disasters. In contrast with other tools developed for post-natural disasters, the PHDS does not disregard sources of distress such as prolonged changes to daily life present after a hurricane which have been found to contribute to the adverse mental health outcomes or exacerbate prevalent cases of depression [21,22,23]. This tool assesses the level of distress of an event occurrence by quantifying the subjective scaled impact on an individual. For each question, participants indicated whether they experienced it or not. If it was experienced by the participant, they subsequently indicated the degree of distress (1–5) that the item/exposure caused them. The PHDS was initially developed in English and was then translated using a forward-backward protocol [24]. The PHDS is freely available in both English and Spanish, without permission for use or modification [12].

#### 2.2.2. GAD-7 Questionnaire

The GAD-7 scale is used as a screening tool in primary care and mental health settings for common anxiety disorders, including generalized anxiety disorder and panic disorder. The GAD-7 uses a 4-point 0 to 4 grading scale and has a reported sensitivity of 89% and specificity of 82% [25].

#### 2.2.3. K6 Questionnaire

The K6 is used as a screening tool for predicting a clinical outcome of severe mood and anxiety disorders. The K6 is a 6-question survey using a 1 to 5 grading scale with a sensitivity of 36% and a specificity of 96% [7,26].

#### 2.2.4. PHQ-9 Questionnaire

The PHQ-9 is used as a screening tool as a first-step assessment in primary care for Major Depressive Disorder (MDD). It is a 9-question survey using a 0 to 3 grading scale with a sensitivity of 88% and a specificity of 88% [27,28].

#### 2.2.5. PCL-5 Questionnaire

The PCL-5 is a 20-item report which assesses the possible presence of post-traumatic stress disorder (PTSD); however, it is not a standalone diagnostic tool. This tool uses a 5-point Likert scale that ranges from 0 to 4 with a sensitivity of 77% and a specificity of 96% [29].

The PHDS, K6, PHQ-9, and the PCL-5 all have published, validated, open-use English and Spanish language versions [12,30,31,32].

### 2.3. Statistical Analysis

Descriptive statistics and receiver operating characteristic (ROC) curves were constructed using R (version 3.6.2) (The R Foundation, Vienna, Austria) [33]. In the event that missing values were encountered upon data review, a process of mean imputation was performed [34]. In cases where more than 5 questions were left unanswered in any questionnaire, the questionnaire was excluded. Although there are many alternatives for determining the cutoff point using ROC curves, we used Youden’s J statistic to derive a single cutoff for the PHDS. Youden’s J statistic is the maximum sum of both sensitivity and specificity, considered an optimum cutoff point for diagnostic tests as it maximizes specificity [35,36,37,38]. To calculate the cutoff points we used the R package “OptimalCutpoints” [36]. To calculate the ROC curve, the package “pROC” was used [37]. In addition to disorder-specific cutoffs, AUC and ROC analyses were completed for each of the four tools by which the PHDS was evaluated (K6, GAD-7, PHDS, and PHQ-9). To evaluate how well the PHDS identifies a participant with a “severe risk of mental health disorder” in any one of the aforementioned assessment tools, a multi-index metric was created for the purpose of defining a single PHDS cutoff relevant to all of the assessment tools used; this metric is a binary measure of positivity in at least one of the assessment tools, thus an individual who scored as positive in any of the previous tools is scored as positive in the multi-index metric.

## 3. Results

### 3.1. Characteristics of the Participants

Of the 872 participants, 200 (23%) returned incomplete packets (more than five blank answers) and were excluded from the analysis, giving a final response rate of 77%. The age of the remaining 672 participants ranged from 18 to 70 years of age, with the mean age equal to 43.5 (SD = 17.8); 33% were men and 65% were women—a significant female predominance (Table 1). Analysis of respondents by gender and group-wise comparison demonstrated no gender difference in the cumulative scores for the PHDS. No significant difference was found between the female and male averages (female mean ± SD, male mean ± SD) for the K6, GAD-7, and the PCL-5, which were (7.32 ± 6.29, 5.73 ± 5.79), (6.05 ± 5.80, 4.73 ± 5.12), and (16.54 ± 16.90, 12.63 ± 17.51), respectively (Table 2)**.** The majority of respondents were white Hispanics; we found no significant difference in assessment scores across race nor ethnicity.

### 3.2. Predictive Accuracy of PHDS and Disorder Cutoffs Indices

Individual ROC analysis of the PHDS in relation to PCL-5, GAD-7, K6, and the PHQ-9 was performed; three of the analyses—K6 being excluded for its homology to the other plots and to maximize legibility—are plotted together in Figure 1. The AUC (95% CI) for the PHQ-9 was 0.84 (0.79–0.90), and for the PCL-5 0.766 (0.71–0.84), while the AUC for the GAD-7 and K6 was 0.753 (0.69–0.79) and 0.744 (0.68–0.80), respectively. These values range from an “acceptable” level of accuracy to a “good” level of accuracy, as in the case of the PHQ-9, for predicting the risk of adverse psychological outcomes [39].

### 3.3. Derivation of Both Disorder-Specific Cutoffs, as Well as a Single PHDS Cutoff Value

The single, general-use cutoff of 41 was derived for the PHDS using an index of positivity across any of the four reference tools. Passing the Youden function on the PHDS versus a binomial variable of positivity across any one of the four mental health assessment tools (GAD-7, K6, PHQ-9, or PCL-5) produced a single multi-index cutoff value for the PHDS. The applicability of this single quantified cutoff of 41 for the PHDS post-disaster research is supported by its superior specificity (0.801), high LR + ratio (2.84), and low FPR (0.20) (Table 3). This PHDS multi-index cutoff shows lower sensitivity (0.56) than the equalizing function-derived cutoff (0.67), but for its intended purpose of defining likely existent mental health disorders in respondents, higher specificity at the expense of sensitivity is supported by seminal studies in the field [7,27,40,41].

Tool-specific Youden cutoffs, wherein each tool corresponds to a distinct mental health disorder; their sensitivities, specificities, and likelihood ratios (LR+) were also estimated. All tool-specific cutoffs were assessed as per the specifications of stringency and specificity put forth by Kroenke et al., [41] namely an LR+ above 2.5 and specificity above 0.75. The minimal likelihood ratio requisite of 2.5 is met by the PHDS versus all four tools (Table 3).

The criterion of Kroenke et al. for a greater than 0.75 specificity [41,42] is met for the PHDS in predicting respondent high-risk status for generalized anxiety, mental distress, and depression; the respective specificities of the PHDS cutoffs for GAD-7, K6, and PHQ-9 are 0.79, 0.82, and 0.90. The concomitant sensitivities of the PHDS cutoffs in predicting GAD-7 (0.59), K6 (0.58), and PHQ-9 (0.63) as being high-risk status are diminished—as expected using the Youden’s index, as opposed to other cutoff derivation indices. The PCL-5 Youden cutoff for the PHDS demonstrated a specificity of 0.70 and thus a false positive error rate (FPR) of 0.30.

### 3.4. Histograms and Distribution Analyses

Comparison of the PHDS score distributions of positive versus negative individuals within the GAD-7, K6, and PHQ-9 show distinct distribution profiles (Figure 2, Figure 3 and Figure 4). Each figure is segregated by the respondent’s risk status per each specific mental health screening tool. In Figure 2, which focuses on GAD risk (defined by GAD-7 scores above 7), a bimodal distribution appears that reflects GAD-7 risk-positive and risk-negative groups.

Similarly, Figure 3 and Figure 4 are corresponding histograms showing the respective PHDS score distributions for both high-risk and non-high-risk respondents to the PCL-5 and the PHQ-9 questionnaires. Superimposed upon each histogram is the corresponding tool-specific PHDS cutoff point. The Youden index cutoffs shown have greater specificity—at the sacrifice of sensitivity—compared with other cutoff point derivation methods which emphasize other criteria (data not shown). Given the design of the PHDS as a first-responder tool, the increased specificity provided by the Youden method is preferred.

## 4. Discussion

The PHDS questionnaire was developed to be used in the early aftermath of natural disasters, as well as the prolonged periods of deprivation and stress that occur in the months following [12]. The PHDS was designed as a metric of trauma and distress and was previously shown to perform better than existing post-disaster assessments, such as the TESS, in terms of its indices of correlation and fit to the GAD-7 and the Peri-traumatic Distress Inventory; analyses of the PHDS’s concurrent and discriminative validity, as well internal reliability, are included therein (IBID).

This study used multiple commonly used clinical assessments that define a person as at risk of specific mental health disorders. The disorder-specific assessment tools that we included in the derivation of the PHDS cutoff are varied and show significant ROC curve homology/overlap. The optimal application of the PHDS in the field—and subsequently the best single cutoff—is dependent upon various factors including the researcher team’s goals and focus, the time post-event, and extenuating circumstances such as availability of relief resources and time for follow-up study and intervention. This study defines an optimal cutoff value for the PHDS of 41, applicable within 12 months of landfall of a natural disaster, in predicting high-risk status across depression, generalized anxiety, and PTSD. Respondents with PHDS scores of 41 and above merit follow-up study and possible intervention as they have increased risk of moderate-to-severe mental health disorders. The evaluation of a cutoff point from a single study is a limitation for multiple reasons. Variation of the impact of environmental disaster on communities, the time of response of local agencies to alleviate stressors, pre-existing conditions, and historical experience with disaster of the community are just a few of the variables that may possibly influence the PHDS scores and the cutoff point, and thus the ability to relate the PHDS to the other indices. However, the AUC for each index were relatively similar and the confidence intervals of the AUC are narrow; thus, it is likely that even under other conditions the cutoff point is likely to be in the same range.

The PHDS, like other mental health analysis tools, including the Peri-traumatic Distress Inventory and the Kessler K6, is not meant to diagnose mental health illness, but rather to define individuals and communities at risk of mental health disorders in the aftermath of natural disasters. Researchers focusing on a specific spectrum of disorders could choose to adopt one of the disorder-specific cutoffs, derived using a single diagnostic tool: the K6, the GAD-7, or PCL-5 (Table 3). Given the existence of various disorder-specific instruments, which most likely show higher specificities and sensitivities for their respective disorders, the goal of the PHDS is not to supplant any of the four diagnostic tools that we have herein used to analyze the PHDS. Rather, it is designed to rapidly identify communities and populations that would merit subsequent mental health follow-up, as well as priority for relief efforts.

Limitations of the current study are varied but can be addressed in follow-up studies and subsequent analyses. Firstly, the PHDS was designed in English, translated to Spanish using a forward-backward protocol and was then validated in both languages [12]; in the current study, the number of respondents who chose the English version was not enough to make any comparisons to the Spanish version. Future implementation and analysis of the PHDS on mainland US would allow further validation of the cutoffs herein presented, in both an English and Spanish language application. Secondly, this validation study occurred solely within a single post-hurricane setting, namely post-Hurricane Maria in Puerto Rico. Cultural and situational differences exist between every disaster and, though the original validation did include a post-Hurricane Harvey population from Texas [12], the present study to define clinically relevant cutoffs occurred solely in Puerto Rico.

Not having the resources to do clinical interviews—the gold standard of mental health diagnosis—is a limitation to studies which require a large sample size; we thus used validated tools rather than diagnostic tools to identify respondents as high-risk for mental health disorders. This imparts a degree of uncertainty to any conclusions as we are using risk assessments, rather than diagnoses, to define cutoffs for the PHDS. A future study that assesses the PHDS’s correlation with actual mental health diagnoses is called for.

This study was completed over a six-month period that began 3 months and ended 12 months after the landfall of Maria, making the optimal time period for assessment post-disaster unclear. We cannot at this point define the best window of time for application of the PHDS, as further analysis, including a time of deployment study, is necessary. Given this study was a follow-up to the original PHDS development study and included time points as far as 12 months after the event, all cutoffs herein derived are presumed valid up to 12 months after the disaster. Future evaluation of the PHDS should include variance of responses as a function of time from the disaster event to confirm the cutoff validity across various time points.

Finally, in favor of a single in-the-field high-risk cutoff value, we have defined a score that was derived from a hybrid measure of high-risk status across any one of three distinct mental health disorders—we acknowledge that this causes a greater degree of uncertainty [43,44]. This uncertainty does err in favor of decreased specificity and given that the PHDS is not meant to diagnose individuals, but rather to identify sub-populations at greater risk for developing mental health disorders, we do not feel it is detrimental to the correct application of the PHDS.

## 5. Conclusions

The valid implementation of the PHDS is post-disaster research, wherein the PHDS demonstrates an individual or community risk for developing various mental health disorders. For this implementation, we calculated the single general risk PHDS cutoff to be 41, using the Youden index to maximize specificity; the sensitivity and specificity of the PHDS in this application are 0.56 and 0.80, respectively. The PHDS can be used directly in the field within one year of disaster occurrence, and can demonstrate a likelihood ratio of 2.84 for respondent positivity across risk of depression, PTSD, and anxiety. Both this and the preceding 2019 validation study demonstrate the PHDS’s utility in identifying high-risk populations based on adverse exposure status and magnitude, thus allowing efficient intervention and resource allocation. Currently the PHDS has been validated in both English and Spanish; further collaborations to increase the PHDS’s availability in other languages is welcome.

## Figures and Tables

**Figure 1 ijerph-19-05204-f001:**
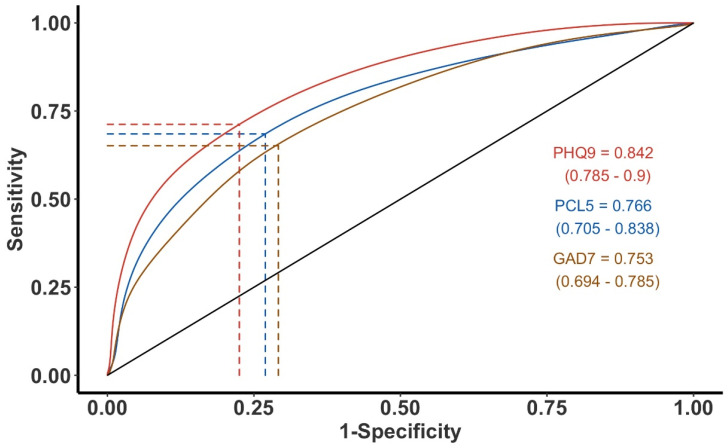
The receiver operating characteristic curve (ROC) for Patient Health Questionnaire 9 (PHQ-9), Post-Traumatic Stress Disorder (PCL-5), and Generalized Anxiety Disorder (GAD-7) and their correlations with Post-Hurricane Distress Scale (PHDS). The K6 ROC plot was homologous to the others and was excluded for legibility. The vertical and horizontal lines are the maximum for the 1-specifity and sensitivity intercepts; the optimal compromise between the specificity and sensitivity defined by Youden’s J index. The values in parenthesis are the 95% confidence intervals of each the Youden’s indices.

**Figure 2 ijerph-19-05204-f002:**
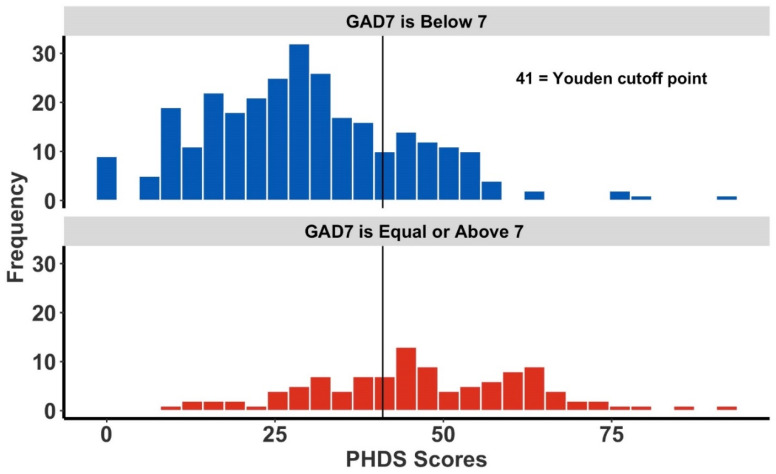
The frequency of Post-Hurricane Distress Scale (PHDS) scores and Generalized Anxiety Disorder (GAD-7) scores. The overlaid vertical line represents the GAD-7-specific Youden cutoff point. The upper figure (in blue) is all respondents scoring below 7 on the GAD-7, while the figure below (in red) represents individuals deemed high-risk for GAD, having a GAD-7 score of 7 or more.

**Figure 3 ijerph-19-05204-f003:**
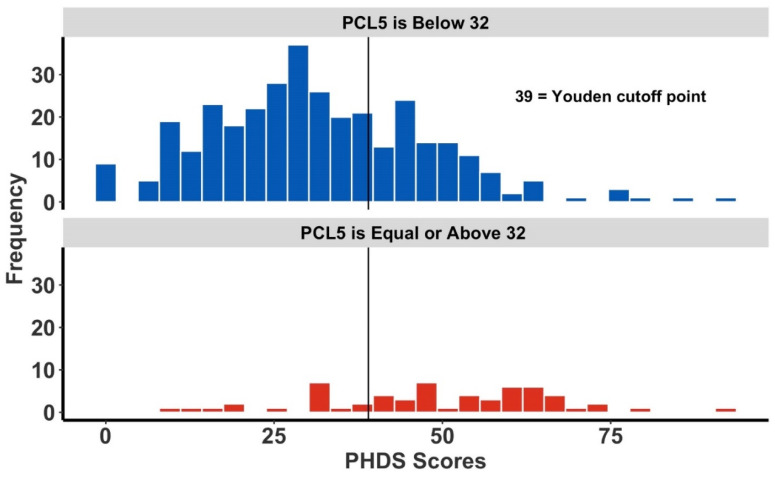
The frequency of Post-Hurricane Distress Scale (PHDS) scores and Post-Traumatic Stress Disorder (PCL-5) scores. The overlaid vertical line represents the PCL-5-specific Youden cutoff point. The upper figure (in blue) is all respondents scoring below 32 on the PCL-5, while the figure below (in red) represents individuals deemed high-risk for PTSD, having a PCL-5 score of 32 or more.

**Figure 4 ijerph-19-05204-f004:**
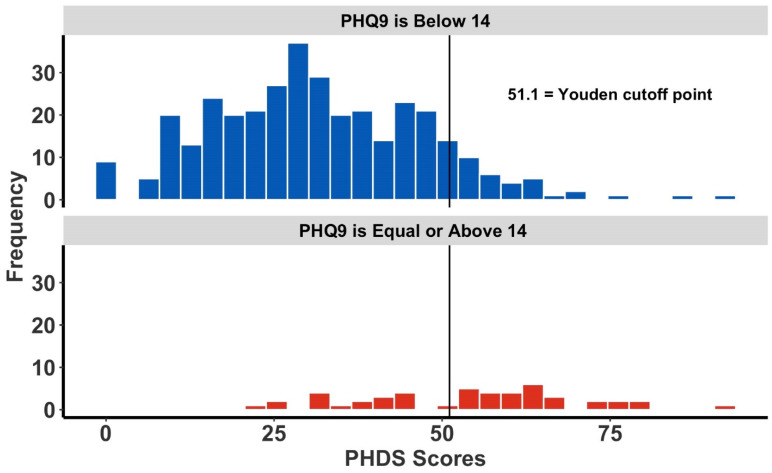
The frequency of Post-Hurricane Distress Scale (PHDS) scores and Patient Health Questionnaire 9 (PHQ-9) scores. The overlaid vertical line represents the PHQ-9-specific Youden cutoff point. The upper figure (in blue) is all respondents scoring below 14 on the PHQ-9, while the figure below (in red) represents individuals deemed high-risk for depression, having a PHQ-9 score of 14 or more.

**Table 1 ijerph-19-05204-t001:** **Participant Demographics and Economic Status (N = 672).** For brevity *Prefer not to answer* is not listed. ^a^
*Married* includes married, living with partner; *Previously Married* includes separated, widowed, divorced. ^b^
*Unemployed* includes full time students, homemakers, looking for employment, not looking for employment. *Employed* includes employed full-time, part-time, and students with part-time employment. ^c^
*Northern region*: Arecibo, Dorado, Florida, Hatillo, Toa Alta, Vega Baja; *Southern region*: Arroyo, Coamo, Guayama, Guayanilla, Juana Diaz, Patillas, Penuelas, Ponce, Salinas, Santa Isabel, Villalba; *Eastern region*: Caguas, Fajardo, Gurabo, Humacao, Juncos, Las Piedras, Maunabo, Naguabo, Rio Grande, San Lorenzo, Vieques, Yabucoa; *Western region*: Cabo Rojo, Hormigueros, Isabela, Lares, Mayaguez, Yauco; *Metropolitan region*: Bayamon, Carolina, Cupey, Guaynabo, San Juan, Trujillo Alto; *Central región*: Aguas Buenas, Aibonito, Barranquitas, Cayey, Ciales, Cidra, Corozal, Jayuya, Morovis, Orocovis, Utuado; *USA***:** California, Connecticut, Florida, Michigan, New Jersey, Texas.

**Age Range (Median)**	18–94 (43.5)				
18–25	**N** 140	**(%)** (21)	35–49	**N** 153	**(%)** (23)
26–34	100 (15)		50+	271	(40)
**Gender**					
Female	434	(65)	Male	221	(33)
**Civil Status ^a^**					
Never married	247	(37)	Previously Married	224	(33)
Married	193	(28)			
**Ethnicity**					
Hispanic	427	(97)	Not Hispanic	4	(1)
**Race**					
White	271	(61)	African American	37	(8)
Native American	28	(6)	Asian	9	(2)
Hawaiian/Pacific islands	3	(1)	Other	80	(18)
**Education Level**					
Less than High School	52	(8)	Bachelor’s or associate degree	247	(36)
High School	133	(20)	Graduate or professional degree	122	(18)
Some college, no degree	111	(17)			
**Employment Status ^b^**					
Employed	292	(49)	Retired/pensioned	87	(15)
Unemployed	176	(30)	Incapacitated	35	(6)
**Combined household income**					
US $10,000 or less	192	(29)	US $26,000 to $49,999	126	(53)
US $10,000 to $25,999	252	(38)	US $50,000 or more	50	(7)
**Current living location ^c^**					
P.R: Eastern region	206	(31)	P.R: Northern region	24	(4)
P.R: Southern region	178	(26)	P.R: Western region	20	(3)
P.R: Central region	145	(22)	USA (emigrated participants)	18	(3)
P.R: Metropolitan region	73	(11)			
**Household Ownership**					
Homeowner	329	(65)	Living with friend or family member (not paying rent)	48	(9)
Rent	108	(21)	Government paid home	6	(1)
Hotel/Motel	3	(1)	Homeless	1	(0)
**Stayed in Puerto Rico During Hurricane Maria**			**Left Puerto Rico** **after Hurricane Maria**		
Yes	647	(96)	Yes	61	(9)
No	23	(3)	No	526	(78)

**Table 2 ijerph-19-05204-t002:** **Comparison of cumulative PHDS, K6, GAD7, PHQ9, PCL5 scores segregated by gender.** Fischer’s Exact test: *p* < 0.05, K6 = Kessler K6, GAD-7 = Generalized Anxiety Disorder, PHQ-9 = Patient Health Questionnaire 9, PCL-5 = PTSD Checklist for DSM V, PHDS = Post Hurricane Distress Scale.

	Female Participants		Male Participants	
	Valid Number of Participants	Mean Score ± SD	Valid Number of Participants	Mean Score ± SD
**PHDS (N = 655)**	648 (66%)	34.40 ± 16.92	333 (34%)	32.19 ± 16.23
**K6 (N = 426)**	270 (63%)	7.32 ± 6.29	156 (37%)	5.73 ± 5.79
**GAD-7 (N = 644)**	425 (66%)	6.05 ± 5.80	219 (34%)	4.73 ± 5.12
**PHQ-9 (N = 425)**	270 (64%)	6.38 ± 6.00	155 (37%)	5.21 ± 5.80
**PCL-5 (N = 422)**	270 (64%)	16.54 ± 16.90	152 (36%)	12.63 ± 17.51

**Table 3 ijerph-19-05204-t003:** PHDS High Risk cut-off selection based on PCL-5, GAD-7, PHQ-9 and Multi-Index Hybrid Receiver Operator Characteristic (ROC) Curve Analyses. Sens, spec = sensitivity, specificity, LR+ = Positive Likelihood Ratio, FP =False Positives, TN = True Negatives, FN = False Negatives, TP = True Positives, FPR = False Positive Rate, FNR = False Negative Rate.

		GAD7	K6	PHQ-9	PCL-5	Hybrid Multi-Index Cutoff
**Youden (Maximising)**	Cut-off	41	46	51	39	41
	sens, spec	0.593, 0.791	0.576, 0.821	0.632, 0.901	0.746, 0.701	0.560, 0.801
	LR+	2.800	3.212	6.395	2.495	2.84
	FPR FP TN	0.210 96 362	0.179 64 293	0.099 37 337	0.299 104 244	0.200 88 353
	FNR TP FN	0.407 105 72	0.424 38 28	0.367 31 18	0.250 44 15	0.440 116 91
**AUC (95% CI)**		0.753 (0.694–0.785)	0.744 (0.675–0.819)	0.842 (0.785–0.900)	0.766 (0.705–0.838)	0.739 (0.698–0.780)

## Data Availability

Not applicable.

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
