# Peer review of "Post-Hurricane Distress Scale (PHDS): Determination of General and Disorder-Specific Cutoff Scores"

_ijerph, 2022, doi:10.3390/ijerph19095204_

Round 1
Reviewer 1 Report
Dear Author(s), congratulations on a fine study, my only suggestion is that given the significant limitations described in the paper that perhaps the conclusion is rather strongly worded in what the tool can achieve and how effective its utility is.
Author Response
Reviewer 1
We appreciate your suggestions and have made the following changes:
Concern: "Perhaps the conclusion is rather strongly worded in what the tool can achieve and how effective its utility is."
Response: We appreciate this suggestion and have removed various superlatives and subjective adjectives as to the utility of the PHDS from the conclusion.
Reviewer 2 Report
It is not clear why specificity and sensitivity are used yet no explanation of tools' validity and reliability psychometrics.
In the current living location, the reader cannot tell if that living location is a result of the hurricane or prior to the hurricane. For example, how many of these participants were relocated due to the hurricane?
Household Ownership? Prior to or post-disaster? How many of the their homes were damaged or destroyed?
Data collection for 3 months to 9-months post-hurricane or 3 months to 12 months? There is a discrepancy between body of paper and abstract.
Author Response
We appreciate your suggestions and have made the following changes:
Concern (1): “It is not clear why specificity and sensitivity are used yet no explanation of tools' validity and reliability psychometrics.”
Response (1): We had previously published the validation statistics for the PHDS in 2019 (PMID: 30841955). We have included a reference to this study in the discussion section of the present submission: “The PHDS was designed as a metric of trauma and distress and was previously shown to perform better than existing post-disaster assessments, such as the TESS, in terms of its’ indices of correlation and fit to the GAD-7 and the Peritraumatic Distress Inventory; analyses to the PHDS’ concurrent and discriminative validity, as well internal reliability, are included therein(IBID).” In the initial validation study included are the following measures of validity and reliability:
- Internal reliability: Cronbach alpha across the 5 factors
- Concurrent Validity: Spearmen rank-order correlation of the 5 factors, R^2, and AIC
- Discriminant validity: K-S. D comparison (non-parametric)
We are unable to include these data as they are already published; the statistics we have included are relevant to the goal of the study, derivation of a cut-off values for a pre-existent, validated tool.
Concern (2): In the current living location, the reader cannot tell if that living location is a result of the hurricane or prior to the hurricane. For example, how many of these participants were relocated due to the hurricane?
Response (2): This is an excellent question. We published a follow-up study in 2020 ( PMID: 31221231) on language as an additional stressor for people who relocated to the continental US after the hurricane. That study focused on those who relocated; this present study excluded those who relocated out of Puerto Rico. A third study, currently underway, focuses on those who lost their home, or were homeless at the time of Hurricane Maria; the distinction of temporary loss of abode, or relocation due to the storm, was not included in this study, as people’s definition of temporary and/or permanent relocation varies significantly.
Finally, the initial validation study (2019, PMID 30841955) did analyze the inclusion of “did you lose your home?” and “did you relocate?” both of which were excluded with principal axis factor analysis, as not relevant to any of the four dimensions of the questionnaire. This was of great surprise to us, but we did comply as our statistical analyses deemed fit.
To address this, we have added a clarification sentence in the methods: “Relocation, whether temporary or permanent, is not included in any of the four factors that comprise the PHDS (PMID: 31221231). A parallel study by our group, currently under review, does assess the contribution of homeless and relocation to mental health outcomes; this study does not make distinction across respondents based on having relocated in the aftermath of the storm.”
Concern(3): Household Ownership? Prior to or post-disaster? How many of their homes were damaged or destroyed?
Response(3): This is a very important contributing factor to mental health outcomes in the aftermath of natural disasters. It is included as a question in the actual PHDS questionnaire (PMID: 31221231). Reviewing responses, we have attempted to answer the questions you posed. These and similar findings will be included in the forthcoming publications, previously referred to in response (2)
A) Household ownership 65.0% (post-disaster respondents)
B) Prior to or post-disaster: I am sorry, we only have post-disaster data; I can, however quote island-wide figures, pre-disaster: 63% (2014 Hinojosa, J)
C) How many homes damaged or destroyed: 65.5% - The PHDS also includes Likert scale for how much said damage impacted the respondent, the mean degree of damage impact was 3.2 on 1-5 Likert scale.
Concern (4): Data collection for 3 months to 9-months post-hurricane or 3 months to 12 months? There is a discrepancy between body of paper and abstract.
Response (4): Thank you for noticing this discrepancy. The period of data collection is correctly written as 3-12 months post hurricane, and has been corrected in all instances in the submission.
Reviewer 3 Report
The manuscript is well-written and discusses the Post Hurricane Distress Scale. The Post-Hurricane Distress Scale (PHDS) has quantitative measures of both acute and prolonged distress, attributable to meteorological and hydrological disasters. This scale represents an interesting and useful instrument for disaster first-responders and researchers. This study aims to determine a cutoff threshold value that best correlates scores with validated mental health risk-assessment instruments. In healthcare, a gap exists between what is known from research and what is practiced. Understanding this gap depends upon our ability to robustly measure research utilization, thus, I would like to congratulate the authors for this scale and cut-offs proposal, which could be very helpful.
Major Comments
The Methods section explain clearly what was done. It was mentioned in the manuscript the Institutional Review Board approval, however the authors did not mention concrete ethical issues (e.g. informed consent, return of results and perhaps therapeutic support for participants who need it, etc.), perhaps it would be important to clarify these aspects. The statistical analysis in this paper is suitable. I suggest the inclusion of one or two more recent references on the subject.
Minor Comments
Table 2 - pay attention to formatting (spacing)
Figure 1 - pay attention to formatting (bold description)
Author Response
Concern (1): “The Methods section explain clearly what was done. It was mentioned in the manuscript the Institutional Review Board approval, however the authors did not mention concrete ethical issues (e.g. informed consent, return of results and perhaps therapeutic support for participants who need it, etc.), perhaps it would be important to clarify these aspects.
Response (1): This is an excellent observation. We had a training session with all of our data collectors (nursing and medical students) and did provide significant counseling when appropriate. To address the concern, we are appending the following explanatory paragraph to this ends, taken from our published pilot study of 2019 (PMID: 31221231):
“All participants were given the purpose of the study by 1 of the researchers, and, if consent was given, the participant was provided with a written consent form that was completed separately to ensure anonymity. Questionnaires were self-administered in the presence of a trained researcher. If, at any point, a participant requested termination of the session, due to emotional distress, the session was ended and participants were informed of psychological resources available to them either through the San Juan Bautista School of Medicine or otherwise available island-wide.”
Concern (2): “The statistical analysis in this paper is suitable. I suggest the inclusion of one or two more recent references on the subject.”
Response (2): We added the following relevant references:
Zou KH, O'Malley AJ, Mauri L. Receiver-operating characteristic analysis for evaluating diagnostic tests and predictive models. Circulation. 2007 Feb 6;115(5):654-7. doi: 10.1161/CIRCULATIONAHA.105.594929. PMID: 17283280.
Ruopp MD, Perkins NJ, Whitcomb BW, Schisterman EF. Youden Index and optimal cut-point estimated from observations affected by a lower limit of detection. Biom J. 2008 Jun;50(3):419-30. doi: 10.1002/bimj.200710415. PMID: 18435502; PMCID: PMC2515362.
Eekhout I, de Vet HC, Twisk JW, Brand JP, de Boer MR, Heymans MW. Missing data in a multi-item instrument were best handled by multiple imputation at the item score level. J Clin Epidemiol. 2014 Mar;67(3):335-42. doi: 10.1016/j.jclinepi.2013.09.009. Epub 2013 Dec 2. PMID: 24291505.
Concern (3): Table 2 - pay attention to formatting (spacing)
Response (3): Thanks for the observation. We fixed the space between the lines so that they were equal.
Concern (4): Figure 1 - pay attention to formatting (bold description)
Response: Thanks for the observation. We bolded the entire descriptive sentence.
Round 2
Reviewer 2 Report
The answers to the reviewer queries strengthen the original submission.